# Design of Multi Standard Near Field Communication Outphasing Transmitter with Modulation Wave Shaping

**Žiga Korošak [1], Nejc Suhadolnik [1] and Anton Pleteršek [2],***

[1]  STMicroelectronics d.o.o., Tehnološki Park 21, 1000 Ljubljana, Slovenia; ziga.korosak@st.com (Ž.K.); nejc.suhadolnik@st.com (N.S.)
[2]  Faculty of Electrical Engineering, University of Ljubljana, Tržaška Cesta 25, 1000 Ljubljana, Slovenia
*   Correspondence: anton.pletersek@fe.uni-lj.si; Tel.: +386-41-961-342

**Abstract:** The aim of this work is to tackle the problem of modulation wave shaping in the field of near field communication (NFC) radio frequency identification (RFID). For this purpose, a high-efficiency transmitter circuit was developed to comply with the strict requirements of the newest EMVCo and NFC Forum specifications for pulse shapes. The proposed circuit uses an outphasing modulator that is based on a digital-to-time converter (DTC). The DTC based outphasing modulator supports amplitude shift keying (ASK) modulation, operates at four times the 13.56 MHz carrier frequency and is made fully differential in order to remove the parasitic phase modulation components. The accompanying transmitter logic includes lookup tables with programmable modulation pulse wave shapes. The modulator solution uses a 64-cell tapped current controlled fully differential delay locked loop (DLL), which produces a 360° delay at 54.24 MHz, and a glitch-free multiplexor to select the individual taps. The outphased output from the modulator is mixed to create an RF pulse width modulated (PWM) output, which drives the antenna. Additionally, this implementation is fully compatible with D-class amplifiers enabling high efficiency. A test circuit of the proposed differential multi-standard reader's transmitter was simulated in 40 nm CMOS technology. Stricter pulse shape requirements were easily satisfied, while achieving an output linearity of 0.2 bits and maximum power consumption under 7.5 mW.

**Keywords:** RFID; EMVCO; NFC; outphasing; PWM; DLL; TX wave shaping

## 1. Introduction

Radio frequency identification (RFID) has gained prominence in recent years as one of the dominant technologies in banking [1,2], access control [3], ticketing [4,5], tracking [6] and in the Internet of things [7]. The technology can broadly be split into devices that operate in the following frequency bands: low frequency (LF), high frequency (HF) and ultra-high frequency (UHF). HF RFID devices that operate at 13.56 MHz are also called near field communication (NFC) devices. These come in two categories. The first category is readers or proximity coupling devices (PCD), which are larger, more complex devices that act as masters in the communication. They transmit a magnetic field that powers the tags or proximity inductively coupled cards (PICC), which are the second category of devices. They are usually simpler devices that can be entirely powered from the reader. They act as slaves in the communication and respond to reader's commands by modulating reader's magnetic field.

Several standards exist in the field. In the HF frequency range, the most prominent standards are ISO/IEC 14443 A/B, ISO/IEC 15693 and JIS: X6319-4/FeliCa [8–10]. A comparison between standards is shown in Table 1.

**Table 1.** Standards comparison. ASK stands for amplitude shift keying, OOK stands for on–off keying, NRZ-L stands for non-return to zero level, BPSK stands for binary phase shift keying.

| Standard | ISO/IEC 15693 | JIS: X6319-4/FeliCa | ISO/IEC 14443 A | ISO/IEC 14443 B |
|---|---|---|---|---|
| Carrier frequency | 13.56 MHz ± 7 kHz | 13.56 MHz ± 50 ppm | 13.56 MHz ± 7 kHz | |
| Subcarrier frequency | 423.75/484.28 kHz | Not used | 847.5 kHz, 1.695 MHz, 3.39 MHz, 6.78 MHz | |
| Data Rate | 6.67 kbps, 26.69 kbps | 212 kbps, 424 kbps | 106 kbps, 212 kbps, 424 kbps, 848 kbps, 1.7 Mbps, 3.4 Mbps, 6.8 Mbps | |
| ASK Modulation index (PCD to PICC) | 10%, 100% | 8–14% | 8–100% | 8–14% |
| Data coding (PCD to PICC) | Pulse position coding | Manchester | Modified Miller, NRZ-L | NRZ-L |
| Modulation type (PICC to PCD) | OOK | OOK | OOK, BPSK | BPSK |
| Data Coding (PICC to PCD) | Manchester | Manchester | Manchester, NRZ-L | NRZ-L |

In addition to the standards there also exist specifications released by organizations and industry consortia. The main organizations are EMVCo [11], which is an organization responsible for payment cards, and NFC Forum [12], which is a technology consortium working to further the use of HF RFID technology, branded under their own name NFC. Both organizations release their own versions of specifications, which include previously mentioned standards but also add their own functionality, certification procedures and requirements.

The recent version of EMVCo specification Level 1 Specifications of Payment Systems Contactless Interface Specification Version 3.0 [13] and Contactless Terminal Level 1 Type Approval Proximity Coupling Devices (PCD) Analogue Test Bench and Test Case Requirements version 3.0a [14], as well as newer versions of the specification, introduce more stringent requirements for modulation pulse shapes. Similarly, the recent version of NFC Forum Analog Technical Specification Version 2.1 [15] and NFC Forum Test Cases for Analog Version 2.1 [16] also specify stricter requirements for pulse shapes. Examples of general on–off keying (OOK) and amplitude shift keying (ASK) modulation pulses are displayed in Figure 1. Each of the specifications has its own definition of the modulation pulse wave shape. Overshoot, undershoot and ringing level are highlighted in the figure. Modulation pulses in NFC are measured by the spy coil, which is present in the RF field alongside the PCD and proximity inductively coupled cards (PICC) coils. The size and shape of the spy coil is specified by each of the specifications.

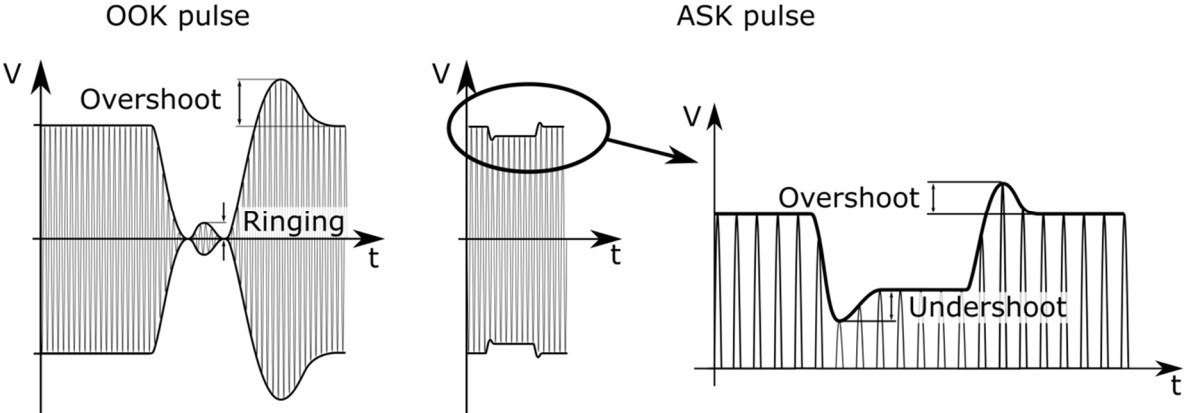

**Figure 1.** Visualization of general purpose OOK and ASK modulation pulses. Overshoot, undershoot and ringing phenomena are highlighted on both types of pulses.

Modulation pulse wave shapes created by the modulator are distorted by the transmission path, which includes an electromagnetic compatibility filter (EMC), matching circuit, antenna coil and proximity coupled devices loading the antenna. Due to magnetic coupling between the PICC and PCD, the quality and resonance frequency of the antenna coil resonance circuit are affected by the antenna load. The shape of the pulses is thus changed with the distance and the number of PICCs present. The most relevant requirement affecting the reader's transmitter is the overshoot and undershoot requirement, which is displayed in Figure 2. The x axis in the figure represents the RF field rise and fall time. EMVCo specifies this requirement to be tested with three reference PICCs, each with two different loads (high linear load (HLZ)—820 Ω and low linear load (LLZ)—333 Ω) and in five different locations. Meanwhile, NFC Forum specifies three of its own reference PICC, all with 820 Ω load in 14 locations. Satisfying the requirements with a modulator that does not enable active wave shaping is extremely difficult.

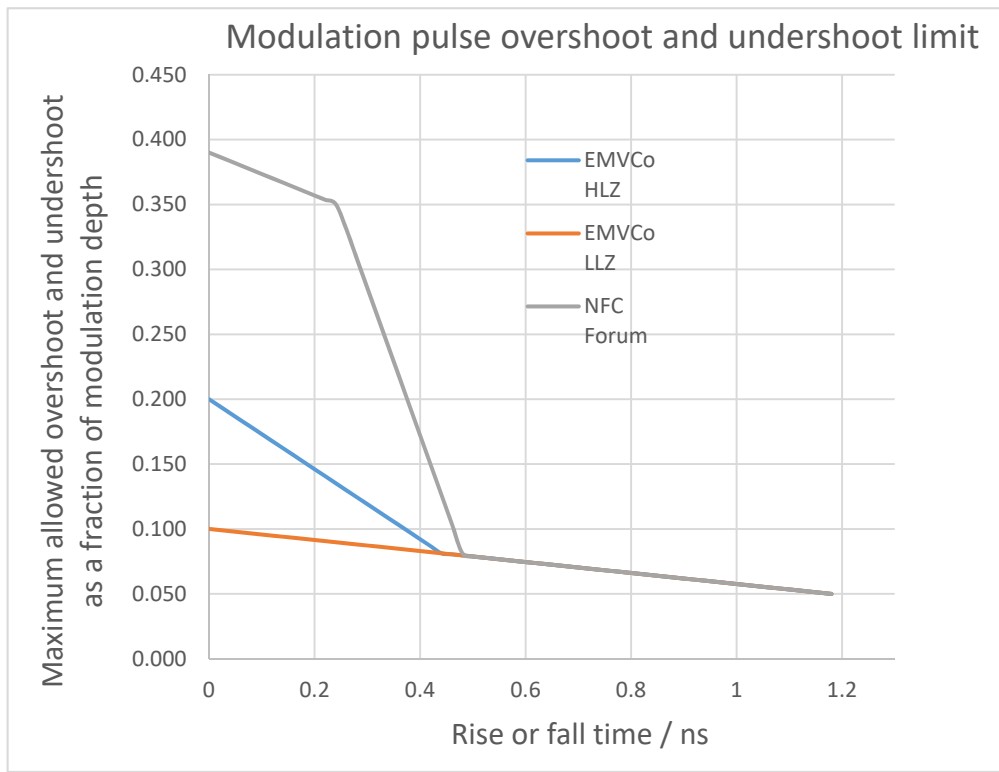

**Figure 2.** Requirement for radio frequency (RF) field overshoot in relation to rise or fall time.

This paper is organized as follows. Section 2 looks at existing works in the field and attempts to expand them to comply with existing new and future specifications. Section 3 proposes a novel solution based on a digital-to-time converter (DTC). Section 4 looks at simulation results for the proposed solution and compares them to similar works. Lastly, conclusions are presented in Section 5.

## 2. Existing Transmitter Solutions for Near Field Devices

In this section, existing works presenting transmitter solutions for near field devices are reviewed. In recent years, not many works have been published in the field of transmitter solutions for NFC. This was due to the simplicity of modulation requirements prior to the introduction of more stringent requirements in the last two years.

### 2.1. Multistandard Polar Transmitter with D-Class Amplifier

The article A multi-standard analog front-end circuit for 13.56 MHz RFID reader [17] presents a transmitter in polar architecture [18], which allows ASK modulation. The pre-

sented transmitter consists of a dual p-stage D-class amplifier without a pre-driver. The dual p-stages can switch between supply voltage and a regulated supply voltage, which is used to create 10% ASK modulation. The regulator can also be switched off to create 100% ASK modulation. The transmitter has three input signals, which are clock, data and type select. The data signal is used to switch between modulated (data = 0) and non-modulated state (data = 1). The type select signal is used to switch between 100% and 10% ASK modulation. When the type select signal is high, the circuit operates in 10% ASK mode. A simplified circuit diagram of a general dual p-stage polar transmitter, which operates similarly to what is presented in Reference [17], is shown in Figure 3.

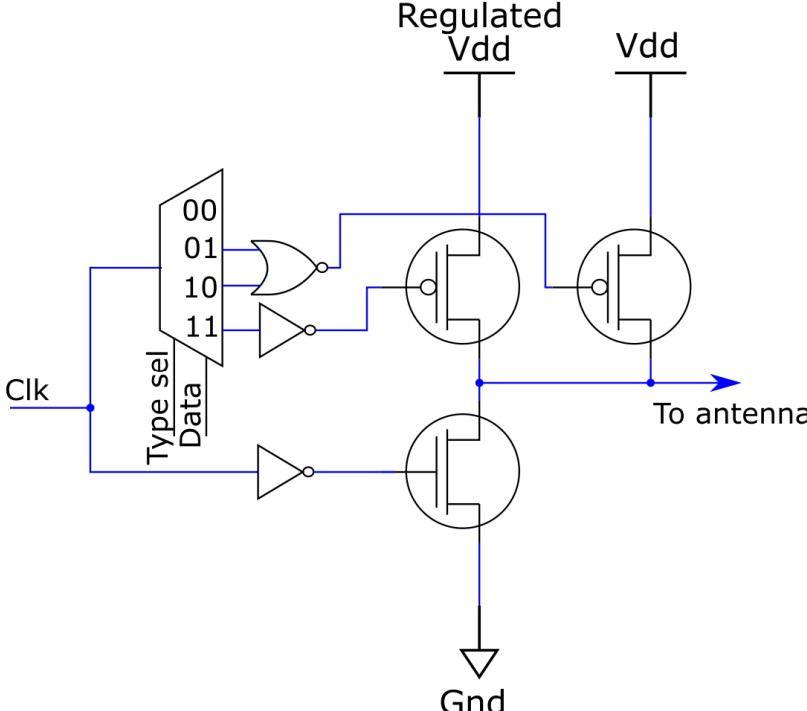

**Figure 3.** Simplified circuit diagram of a general dual p-stage polar transmitter. The Type sel signal controls the mode of operation and can select between ASK and OOK operation. The Data signal controls the modulation state and can switch between modulated and non-modulated states.

There are several drawbacks to the solution [17]. There is no non-overlap control circuit on the D-class amplifier, causing cross conduction [19], which decreases efficiency of the driver, causes aging of the power transistors and limits the scalability of the driver to higher output powers. The solution passes the pulse-shape requirements of the older standards; however, reaching compatibility with the new specifications would be extremely difficult. The pulse shape is entirely dependent on the transmission path, and the solution lacks any means of actively changing the antenna characteristic or the shape of the pulses.

A possible way to expand this work to comply with new standards is by adding adaptive antenna tuning [20]. This can be implemented in a way to ensure a constant magnetic field or constant matching impedance. No research has been done up to date in using automatic antenna tuning to control modulation wave shape. An alternative way to expand this work is to replace the simple regulator used to make ASK modulation with RF digital-to-analog converter. This way, the modulator bandwidth could be high enough to shape the relatively slow modulation pulses.

### 2.2. Multistandard Pulse Width Modulated (PWM) Transmitter Based on Delay Locked Loop (DLL) Architecture

The article Design of 13.56 MHz ASK transmitter for near field communication using a DLL architecture [21] presents an NFC outphasing PWM [18] transmitter based on a DLL

structure. Outphasing is a method of modulation where two transmitted carrier signals' phase relationship is changed in order to create amplitude and phase modulation. Let us assume, for the purpose of explanation, that we have two ideal cosine transmitters of which the phase can be modulated. Transmitted carrier is thus equal to:

$$v(t) = \frac{1}{2}(\cos(2\pi f + \Phi_1(t)) + \cos(2\pi f + \Phi_2(t))) \tag{1}$$

The outphasing angles can be split into a common part, where both transmitters are phase modulated in the same direction, and differential part, where transmitters are phase modulated in opposite directions.

$$\Phi_1(t) = \varphi(t) + \vartheta(t) \tag{2}$$

$$\Phi_2(t) = \varphi(t) - \vartheta(t) \tag{3}$$

Inserting this into a previous equation and applying trigonometric relation for cos(*a* + *b*) we obtain:

$$\begin{aligned} v(t) &= \tfrac{1}{2}(\cos(2\pi f + \varphi(t) + \vartheta(t)) + \cos(2\pi f + \varphi(t) - \vartheta(t))) \\ &= \cos(\vartheta(t)) * \cos(2\pi f + \varphi(t)) \end{aligned} \tag{4}$$

$$\vartheta(t) \in \left[0, \frac{\pi}{2}\right], \; \varphi(t) \in [0, 2\pi] \tag{5}$$

Thus, we obtained the amplitude component of modulation $A(t) = \cos(\vartheta(t))$ and phase component of modulation $\varphi(t)$ by using just phase modulators. The same principle works for square signals as well, with higher harmonics being modulated as well. In the case where both carriers are modulated, differential outphasing is obtained, in which phase modulation and amplitude modulation can be separated. If only one of the carriers is modulated, while the other is left at constant phase, single ended outphasing is obtained. In this case both phase modulation and amplitude modulation happen at the same time and cannot be separated. Carriers can also be mixed prior to amplification. This creates RF PWM signals.

The presented solution is multi standard and supports modulation indexes from 0% to 100%. It is composed of a load-controlled delay locked loop, which locks its output clock to its input clock. In this way a known phase delay is created. The loop has a phase-frequency detector, which compares its input and output phases to create an input signal for the charge pump. The charge pump moves charge to create a control voltage, which is filtered by a first order loop filter. Variable phase delay in the loop is achieved by using a 6-bit controllable offset current in the charge pump. The outphased clock, which is created by the DLL, is mixed with the reference clock in the control stage to create an RF PWM signal. The control stage also functions as a switch between the reference clock and RF PWM clock to create a modulated and non-modulated state. The solution also includes a power stage, which is a D-Class amplifier with a pre-driver. The power stage has 6-bit adjustable output power. A block of a generalized ASK RF PWM transmitter, similar to what is presented in [21], is shown in Figure 4. Arbitrary clock phase generation can be implemented with a DLL, as presented in [21], but phase locked loop implementation is also viable. Both implementations can perform single ended or differential outphasing, the difference between which we tested in Figure 5. The amplifier can be of any kind, but D-class and E-class amplifiers are generally used with this configuration.

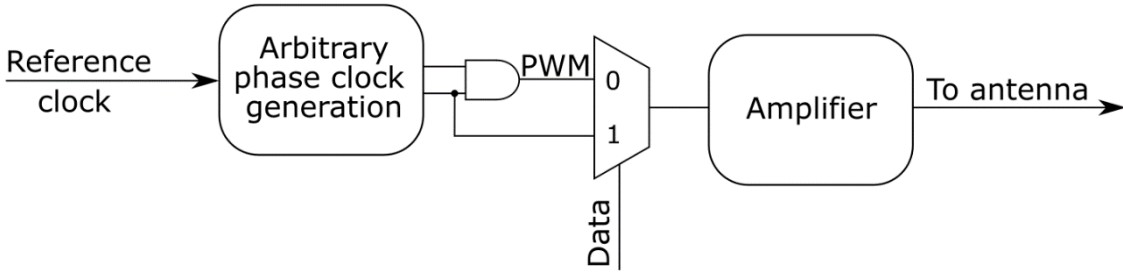

**Figure 4.** Block diagram of a generalized ASK RF- Pulse Width Modulated (PWM) transmitter.

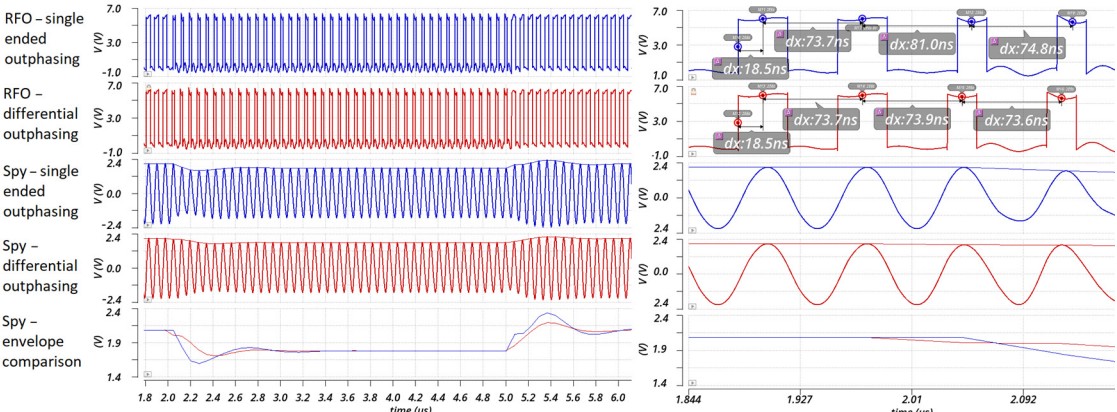

**Figure 5.** Comparison of RF PWM generated by single ended and differential outphasing. Blue lines represent signal generated in single ended outphasing; red lines represent signal generated with differential outphasing.

The largest drawback of this solution is that its outphasing modulation is single ended, which creates parasitic phase modulation along with amplitude modulation [22]. Transmitted phase during the modulated phase is thus dependent on the set ASK index. Additionally, when the transition happens between modulated and non-modulated state for 8–30% ASK, an abrupt change in phase also occurs, which the antenna circuit can translate into an increase of overshoot and undershoot of the envelope, which, in turn, can cause a failure to meet specifications. Figure 5 shows a comparison between an ASK pulse generated with single-ended outphasing (blue) and with differential outphasing (red) for the circuit in Figure 4. In the simulation, a generic single ended antenna with an EMC filter and matching circuit is used along with an ideal D-class amplifier with 0.5 Ω output resistance. The antenna is loaded by an NFC Forum reference listener 6 with a coupling factor of 0.2. Traces shown are from the spy coil and RF output (RFO) of the amplifier. From the envelope comparison it can be observed that single ended outphasing causes a larger undershoot and overshoot. The phase difference caused by the parasitic phase modulation can be seen in the zoomed-in view by observing the length of different periods at the point where the pulse width transition occurs.

### 3. Novel Solution Based on Digital to Time Converter

Work done in [23] was used as a basis for an outphasing transmitter based on a digital-to-time converter in our work. The benefit of this implementation is that it can be made fully digital and can thus take full advantage of advanced process nodes. Outphasing is made fully differential to remove the parasitic phase modulation component and have only ASK modulation. Additionally, this implementation is fully compatible with D-class amplifiers, enabling high efficiency.

The solution we propose for NFC outphasing transmitter consists of transmitter logic with a programmable lookup table, where modulation pulse wave shapes are stored, and a modulator that performs modulation using the wave shape data from the lookup

table. The modulator, operating at four times the carrier frequency, is followed by a shift register based clock divider with a fixed phase offset to achieve 180° phase offset in a non-modulated state. After that, outphased carriers are converted to RF PWM in a digital mixer composed of matched delay "AND" and "NOR" gates. Both gates have one of their inputs negated. Lastly, this solution needs a D-class amplifier to operate, but this amplifier was not included in this work. The block diagram of the proposed solution is shown in Figure 6.

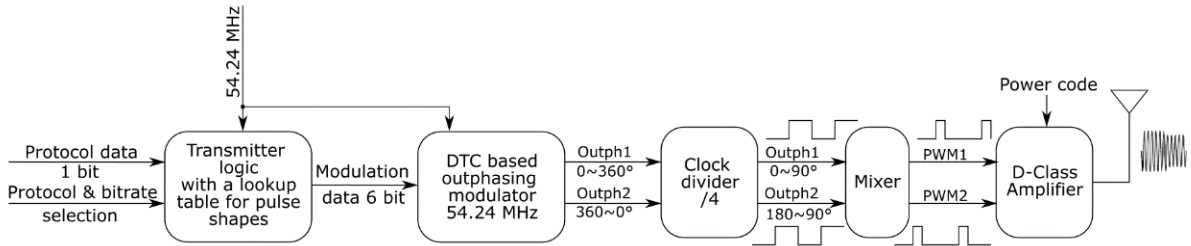

**Figure 6.** Block diagram of the proposed solution. Abbreviations Outph1/2 refer to outphased clock 1/2.

The core of the solution is DTC based outphasing modulator that operates at four times the carrier frequency and allows ASK modulation. Its block diagram is shown in Figure 7. The modulator operation is based on switching between the taps of the delay line to achieve the desired phase on the output. The current controlled delay line together with the charge pump, phase detector and $\pi$ loop filter constitutes a delay locked loop.

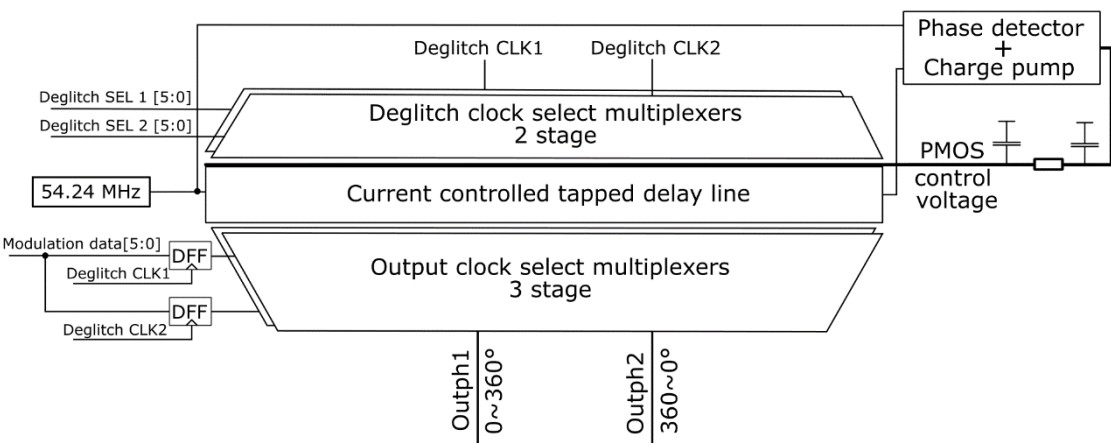

**Figure 7.** Block diagram of a digital-to-time converter (DTC) based outphasing modulator. Abbreviations Outph1/2 refer to outphased clock 1/2. Abbreviation DFF stands for D-flip-flop.

### 3.1. Delay Locked Loop (DLL) Architecture

The solution uses a 64-cell tapped DLL, which produces 360° delay at 54.24 MHz. When frequency is divided down to 13.56 MHz this delay is divided as well to get 90°. Each cell of the delay line produces 280 ps of delay. Using a higher clock allows for smaller delay line size as fewer delay cells are needed and only slightly increases power consumption. A total of 128 noninverting delay cells would be required to achieve differential outputs of −90~0° and 0~90° at 13.56 MHz frequency. Total delay of the delay line would thus be 180°, which would not enable locking of the delay line independent of input clock duty cycle. At four times the base frequency, full 360° lock can be achieved with only 64 cells. A further optimization can be made by using inverters instead of buffers as delay cells, further reducing their size by half. This has an effect where a 180° phase difference is created between odd and even cells, which has to be accounted for with multiplexer

connections. The total number of cells must remain even to ensure an input clock duty cycle independent lock. With higher frequency, the size of the loop filter is reduced as well. Differential modulation is achieved by having separate multiplexers but reusing the same taps for both outputs. Output 1 moves from 0 to 360° and output 2 moves from 360 to 0°. Because the delay cells are inverting, phases 0~180° are available from even taps and phases 180~360° are available from odd taps. Additional differential feedbacks are added between cells to reduce mismatch of the delay and duty cycle between inverting and noninverting taps.

There exist several types of delay cells. For this work, current controlled (Figure 8a) and load controlled (Figure 8b) delay cells were considered. In a current controlled delay cell, current of delay inverters is regulated to achieve a desired delay. A serial resistance of a load capacitor is regulated to ensure a correct delay in load-controlled cells.

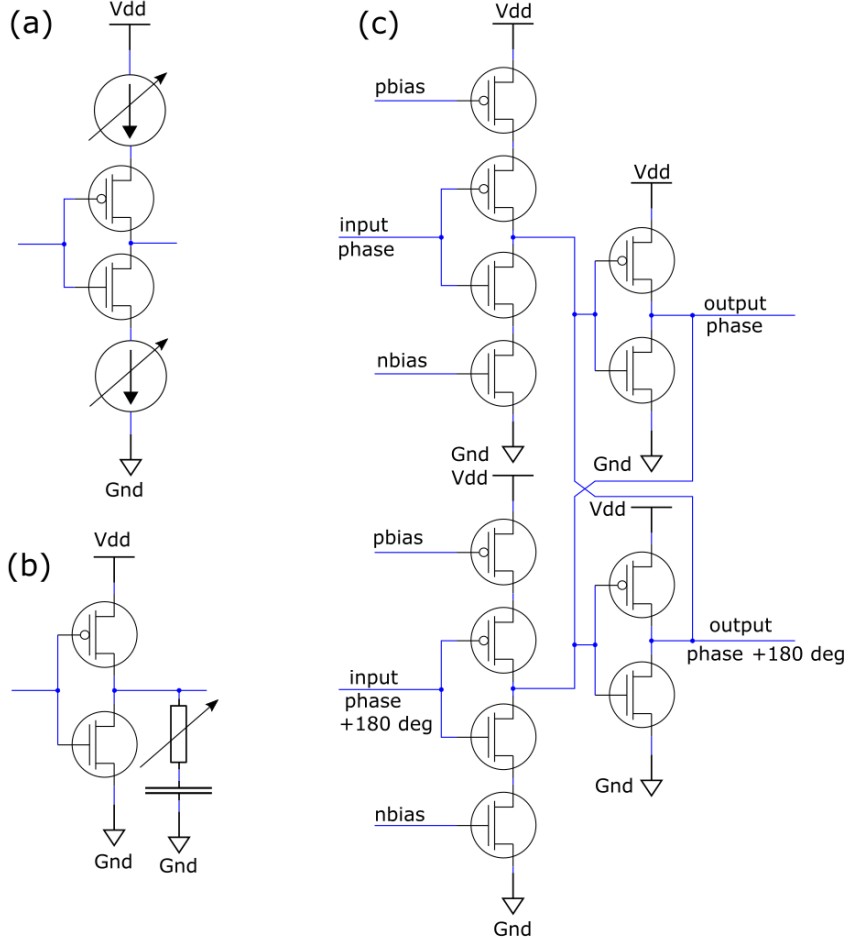

**Figure 8.** (**a**) Current controlled delay cell. (**b**) Load controlled delay cell. (**c**) Circuit diagram of a differential current controlled delay cell with feedback. All bulk connections of transistors in depicted circuits are connected to supply and ground for PMOS and NMOS transistors, respectively. Bar symbols represent supply voltage Vdd, which is equal to 2.4 V.

Current controlled delay cells were found to have higher phase amplification and lower mismatch as well as much lower power consumption in the equivalent area as compared to load-controlled delay cells. The compromise for this is a less linear delay characteristic of the current controlled delay cells. This drawback of current controlled delay cells could be alleviated by adding diodes in parallel to control transistors at the cost of decreased phase amplification. A comparison of delay characteristic of the two cells made from 2.5 V transistors in 40 nm CMOS is displayed in Figure 9. Red lines in Figure 9 represent a load-controlled delay cell, blue lines represent a current controlled delay cell.

On the x axis we have the PMOS control voltage, while the NMOS control voltage is generated from the PMOS control voltage by means of current mirroring. The first segment in the figure represents the phase change of the delayed clock at a given control voltage or phase amplification of the delay cell. The second segment represents the actual delay. It can be observed in Figure 9 that the usable control voltage range of current controlled delay cell is smaller (0.8~1.4 V), but is more centered around half of the supply voltage (1.2 V). A useful range of load-controlled delay cell is higher (0.95~1.8 V) but is more skewed towards higher voltages.

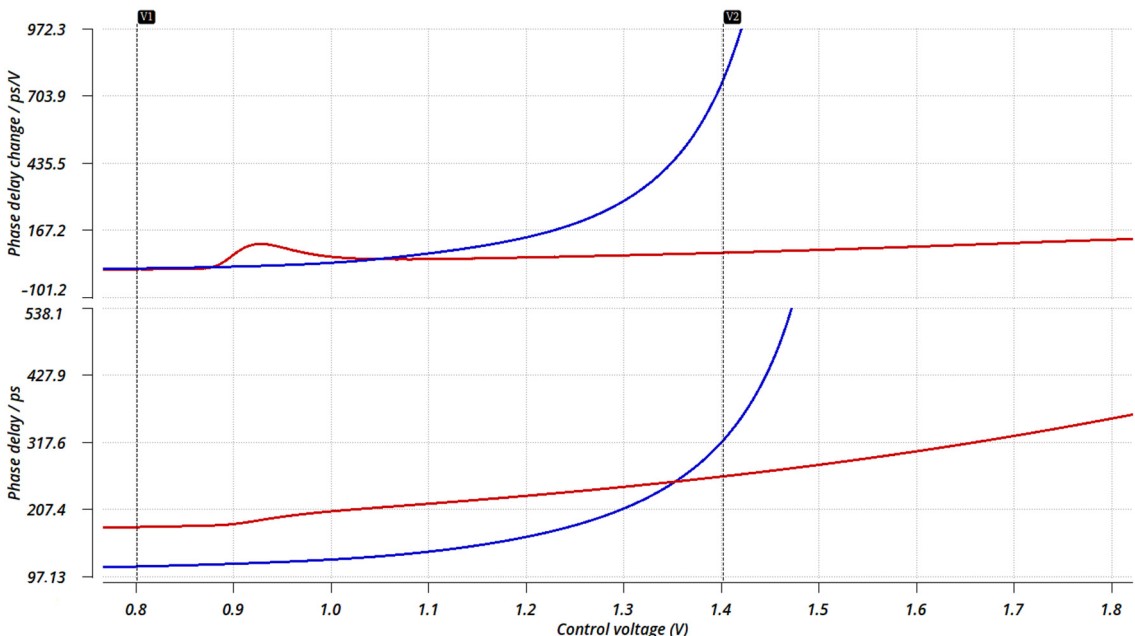

**Figure 9.** A comparison of delay cells. The red lines represent a load-controlled delay cell, blue lines represent a current controlled delay cell.

Current controlled delay cells in differential configuration, as shown in Figure 8c, were chosen for this work. The benefit of using inverters instead of differential amplifier stages to create differential delay cells is a rail to rail operation independent of the control voltage. Delay cells are current controlled on both sides to ensure symmetrical rising and falling edges, as both are effectively used. Differential feedback inverters are sized to have one-fifth to one-third of the current capability of the main delay inverters. The delay line is connected in such a way that the 180° delayed clock is wrapped back to the differential input, as displayed in Figure 10. This enables the delay line to have 360° of total delay with inverting delay cells while keeping a proper delay between inverted and noninverted outputs. The area of the delay line is thus reduced by half, but a nonlinearity is introduced at the point where it wraps back. This can be seen at code 32 in the differential nonlinearity plot in Section 4. To avoid this issue the delay line would either have to be made of noninverting delay cells or be made fully differential with inverting delay cells both of which would double the area. The PMOS control voltage is generated by a charge pump and is filtered by the loop filter as shown in Figure 7. NMOS control voltage is generated by mirroring the PMOS control voltage.

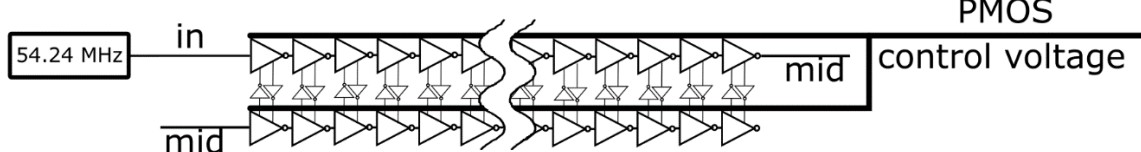

**Figure 10.** Delay line architecture. The number of differential delay stages is 32, totaling 64 delay inverters and 64 cross feedback inverters.

To lock the DLL a single-ended charge pump is used, adapted from the differential charge pump presented in [24]. In front of the charge pump dynamic D-flip-flops (DFF) are used as phase detectors, as shown in Figure 11. The loop filter is of the second order, which improves the phase noise characteristic of the loop [25] from high pass to band pass. Filter in π topology was found to be most appropriate for the application. The first capacitor on the charge pump output creates the first pole while the combination of the resistor and second capacitor creates an additional pole. The second pole is sized to be much weaker than the first pole, to keep the loop overdamped.

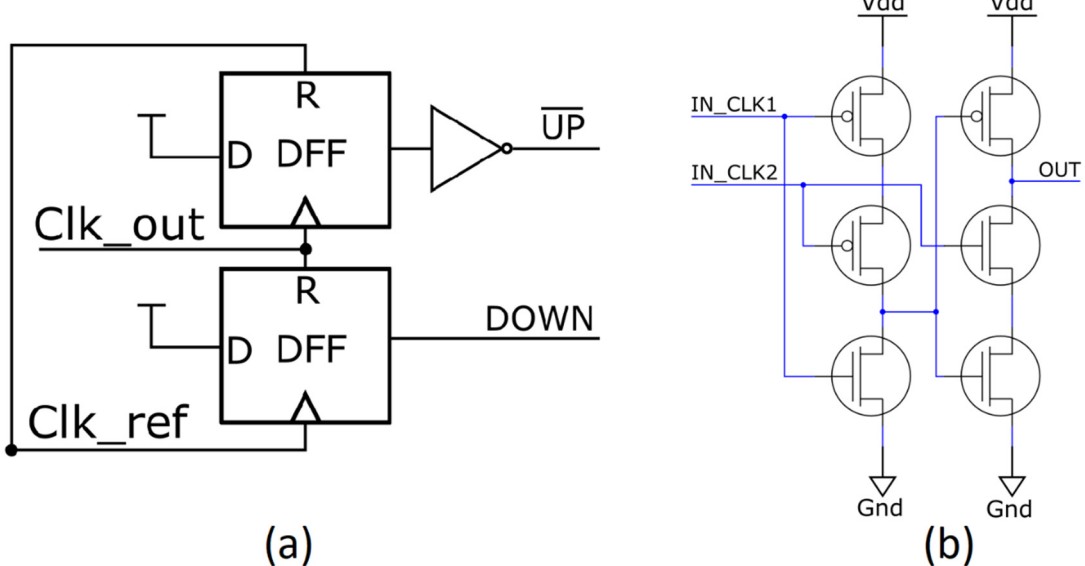

**Figure 11.** (**a**) Phase detector block diagram. (**b**) Dynamic flip flop circuit diagram. Vdd is equal to 2.4 V. All bulk connections of transistors in depicted circuits are connected to supply and ground for PMOS and NMOS transistors, respectively.

### 3.2. Modulator Architecture

Besides the DLL, the second major component of the modulator is its phase multiplexing system, as displayed in Figure 7. This switching system has two subsystems: a deglitching system and an output switching system. The output system switches between available phases created by the DLL to perform the desired phase modulation. This system consists of three stages of four to one multiplexers. The first stage has 16 multiplexers and is common for both outputs. This is an area optimization enabled by the fact that both outputs are never connected to the same first stage multiplexer in differential amplitude only outphasing. The second and third stages of multiplexers are separated for the two outputs, adding an additional 10 multiplexers to the design.

The second part of the switching system is the deglitching subsystem. This is required because glitches on the output of the modulator are not tolerated by the clock divider. Glitches occur when switching to a different phase clock when the current clock and target clock are in different states, as shown in Figure 12.

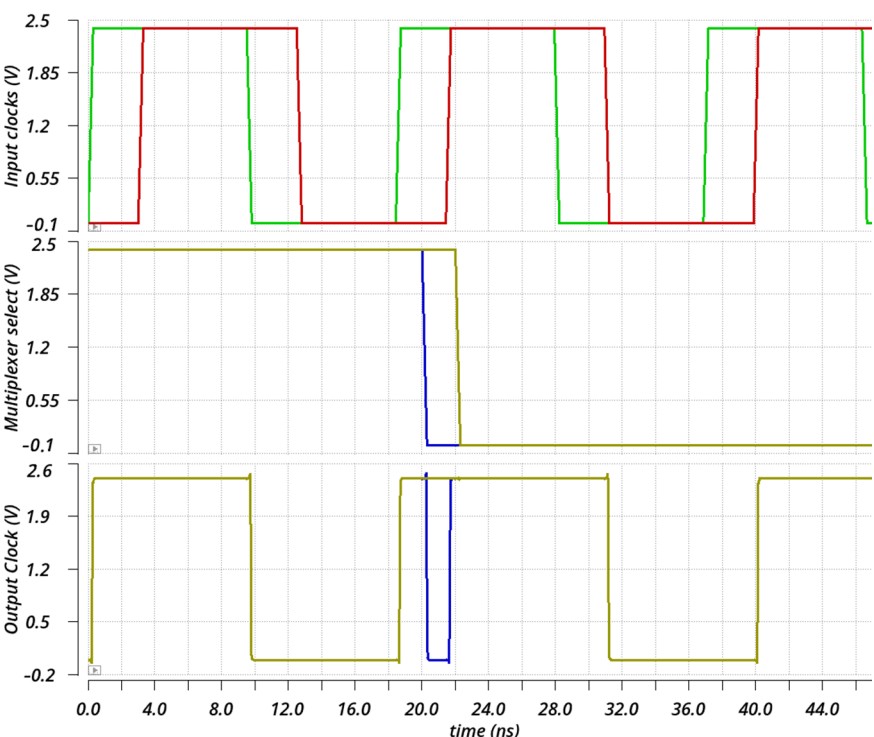

**Figure 12.** Comparison of the switching phase of a multiplexer clock when both clocks are in the same state (dark yellow trace) and in a different state (blue trace), resulting in a glitch. Red and green traces are input clocks, between which the switch happens.

The time interval when both signals are in the same state (high or low) decreases with a phase difference between the signals. This interval reaches zero when the phase difference becomes 180°. When a multi-stage mux is used for output switching this problem becomes even worse, because not all stages of the mux will switch at the same time. If any of the transient states is different than the final state, this will be seen as a glitch at the output. Thus, switching can only happen where none of the transient phases will be in a different state than the current state. Several solutions have been proposed for this problem. Reference [26] proposes a make-before-break system implemented by using several taps at the same time and a 37.5% duty cycle clock. Their solution allows phase to switch for up to 90° per cycle and allows unwrapping of phase. An alternative solution to this problem is to use a dedicated adjustable phase clock for switching the output. This ensures that the switch always happens at the appropriate moment in order not to cause a glitch.

In this work, a dedicated switching clock was implemented for each output in conjunction with adaptive switching of inactive outputs of second stage multiplexers. Such switching increases the size of the time window to switch the output clock without glitches. The switching clock is implemented with an additional multiplexer in the DLL, which only uses five of the six available input bits. The most significant bit is ignored because the overlap of phases where the output can be switched without glitches happens twice per period. This is made by using the same clock phase to switch phases 180° apart. The phase of the switching clock is selected depending on the phase of the output clock in the next cycle. A comparator on the input determines the direction of change from the current cycle to the next cycle. A block diagram of the deglitching system is shown in Figure 13. Connections between the tapped DLL and multiplexer stages are not drawn to avoid unnecessary clutter in the figure. Connections between these elements are made in such a way that modulation setting 0 creates pulses with zero duty ratio and modulation setting 63 creates pulses with 50% duty ratio.

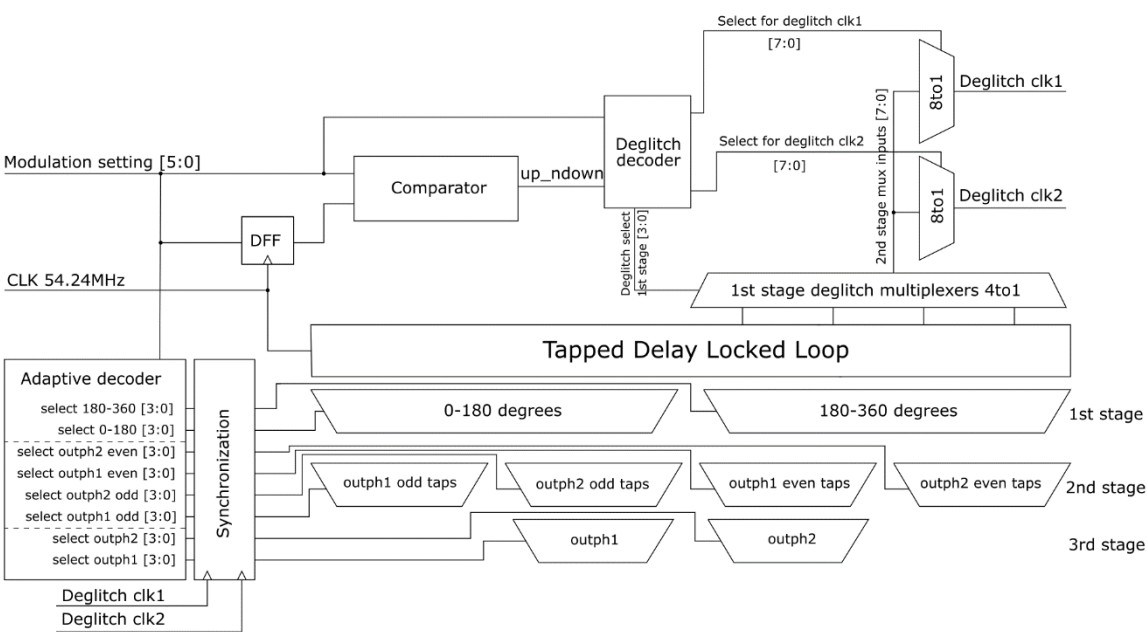

**Figure 13.** Block diagram of the output and deglitching systems.

The implemented system allows outputs to be switched without glitches for 70° each, every 54.24 MHz clock cycle. The modulator input setting is also changed with 54.24 MHz by interpolating intermediate states. This has the benefit of increasing phase change to a theoretical limit of 70° for each output and for every clock cycle of 13.56 MHz. The actual change allowed by interpolation is lower than the theoretical limit due to simplified implementation of interpolation. Actual available phase change is 45° for every output and for every clock cycle. This is more than enough for the intended usage, as the intention is to transition into full modulation depth gradually to shape modulation pulses.

## 4. Results

The presented solution was simulated with a generic differential NFC controller antenna composed of an antenna coil, resonance capacitors, matching capacitors, and EMI filter inductively coupled to an EMV—TEST PICC 2 to guarantee the solution meets specification requirements. Because the modulator has a much higher bandwidth and resolution than the communication protocol data rate, the shapes of the modulation pulses can be actively controlled. Furthermore, when changing the pulse width more gradually at the output, aliasing into the transmission band is reduced due to the zero order hold effect in combination with a square carrier shape [22]. Power consumption of the presented system without the D-class amplifier in the worst-power corner is below 7.5 mW, an amount which is negligible compared to transmitter output power, which is over 1 W for most high end devices on the market today [27–30]. Power consumption could further be drastically reduced by using low voltage transistors instead of 2.5 V transistors.

### 4.1. Output Characteristic

Results in Figure 14 are obtained using an ideal D-class amplifier with 1 Ω output resistance in the proposed solution, simulated over temperatures from −30 to 125 °C and supply voltage variation from 2.3 V to 2.5 V with Monte Carlo local variations. The coupling factor between the antenna and PICC was 0.20, which represents 10 mm distance between the PICC and transmitter antenna. Pulse width in the simulation was swept from input setting 0 to setting 63. Setting 0 is missing from the figure because its duty cycle was 0 and could not be sampled. The first set of lines of the graph represents differential nonlinearity (DNL) of the output duty cycle. The offset generated by the modulator is half a bit, but this is negated by the mixer, which swallows such short pulses. The gain error is

one bit over the entire range of input codes, causing the highest code to be half a bit below nominal. This is visible in the DNL plot in Figure 14. Gain and offset errors are caused by the architecture, which was optimized for area and power consumption. The DNL plot shows a consistent nonlinearity in the middle of the range and at the edges. The middle nonlinearity is up to 0.2 bits in size and is partially caused by the multiplexer architecture where the first stage of multiplexers is common for both outputs and partially by folding of the delay line. Edge errors (left edge is not shown because pulse is swallowed) are caused by changing of the delay at the ends of the delay line. Additional dummy delay stages were implemented at the edge, but they are not differential and could not be perfectly matched to delay stages in the DLL. The amplitude of this error is under 0.3 bits. Both errors can easily be accounted for in the lookup table and are thus not a problem. Total area of the transmitter is ~0.025 mm². We estimate the area would increase by about 15% without the area optimizations while the power consumption would increase by 40%

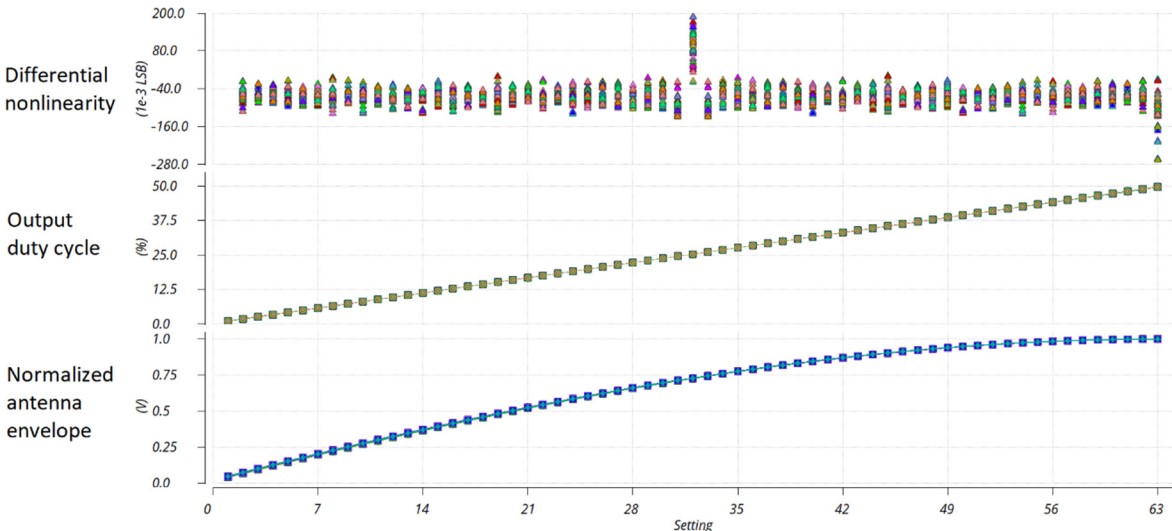

**Figure 14.** Simulation of output characteristic showing DNL, output duty cycle and normalized antenna envelope.

The second set of lines is the sampled duty cycle at the output of the modulator, where good matching between simulation runs can be seen. The third set of lines are the sampled amplitudes of the RF field envelope as seen by the spy coil. The sine transformation characteristic of differential outphasing is also clearly visible in this set of lines.

*4.2. ISO14443 A and B Modulation Pulse Characteristic.*

ISO14443 A and B modulation simulation results in Figures 15–17 are obtained in a nominal corner with generic differential 14 Ω PCD antenna coupled with EMVCO Reference PICC 2. The load on PICC is set to HLZ—820 Ω. Antenna coupling factors used in the simulation are modeled for PICC distances 0–50 mm from the reader antenna as specified by EMVCo. Only results for 0 mm distance are shown, as this antenna exhibits the most challenging behavior at this distance. The point where the antenna performs the worst depends on how the antenna is configured (matching circuit, etc.) and can be different for different antennas, but this behavior cannot be avoided in general. Devices that do not address this issue can exhibit a dead zone in communication, which is undesirable.

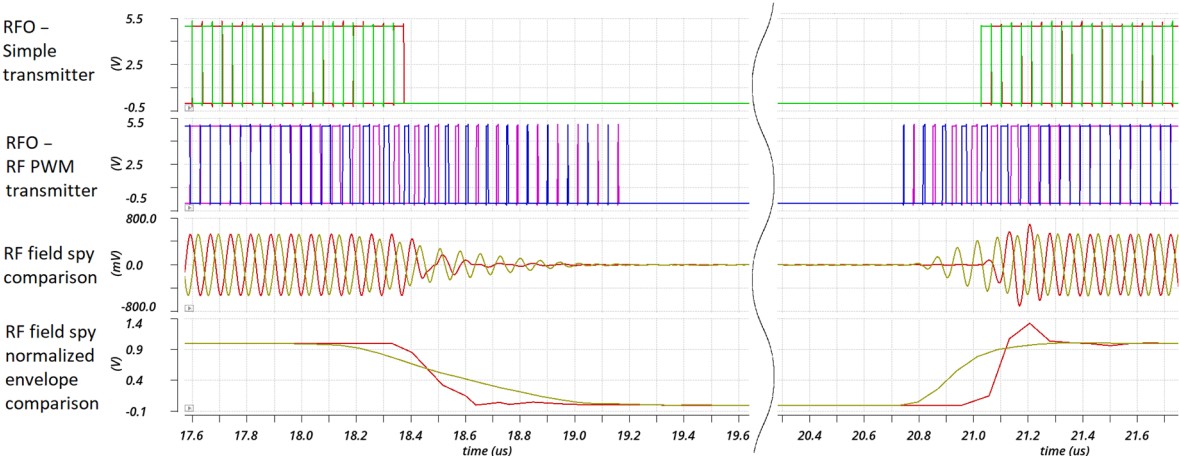

**Figure 15.** ISO14443 type A 106 kbit/s modulation pulse comparison. Yellow line on RF field spy is from the RF PWM transmitter, red line is from the simple polar transmitter. Yellow and purple lines are RF PWM transmitter output clocks. Red and green traces in the first segment of the figure are simple transmitter output clocks.

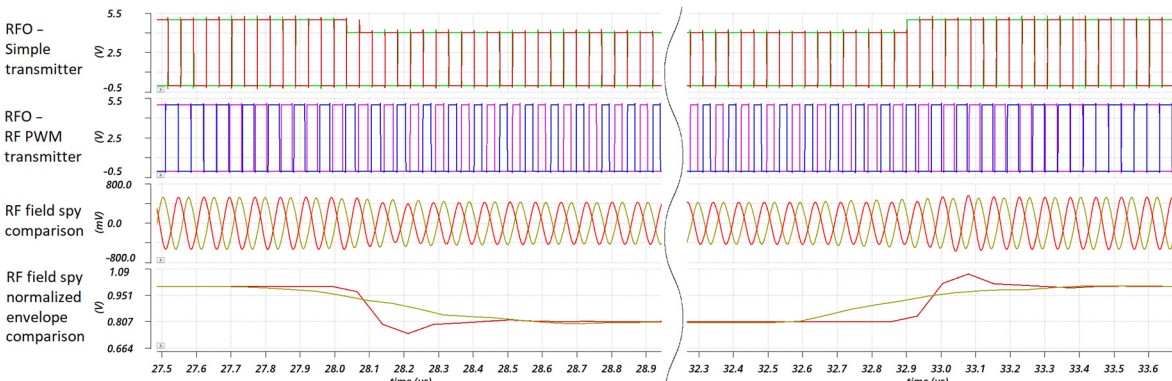

**Figure 16.** ISO14443 type B 106 kbit/s modulation pulse comparison. Yellow line on spy RF field is from the RF PWM transmitter, red line is from the simple polar transmitter.

In Figure 15 a comparison is made between a simple polar transmitter and the solution proposed in this work for an ISO14443 type A 106 kbits/s modulation pulse. The first set of lines in the figure are the RF outputs from the simple polar transmitter. The second set of lines are the RF outputs from the solution presented in this work. The third set of lines is a comparison between the RF field spies for this work (yellow line) and from the simple polar transmitter (red line). The two transmitters are transmitting with different phases for better visualization. The last set of lines is a comparison between normalized RF field spy envelopes. Colors are the same as before. From the figure we can observe that the simple polar transmitter does not comply with the overshoot requirement, but the proposed solution does without significant overshoot with the use of wave shaping. Furthermore, the transition into modulated state with proposed solution is much cleaner, reducing emissions outside the transmission band. Our wave shaping solution allows us to further adjust the shape of rising and falling edges in a programmable lookup table to suit different antennas and the requirements of different protocols and data rates.

The same comparison as before is made for a ISO14443 type B 106 kbit/s pulse and is shown in Figure 16. Again, the simple polar transmitter does not reach the overshoot and undershoot requirement at the worst point required by the EMVCo specification, while our wave shaping solution satisfies the requirements without issue.

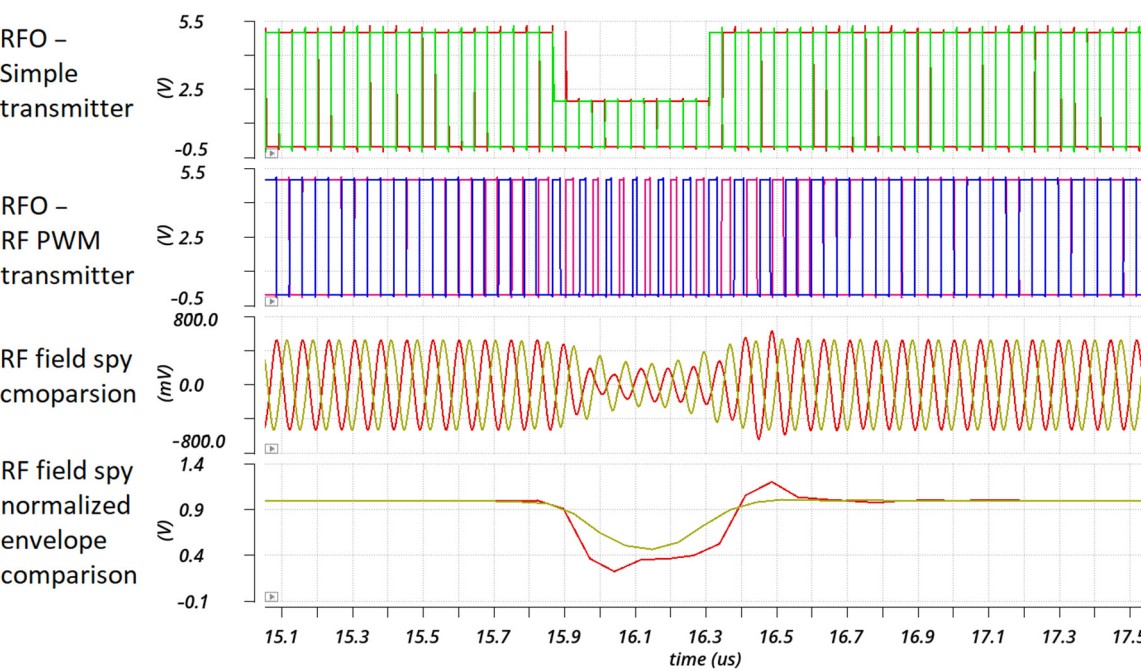

**Figure 17.** ISO14443 type A 848 kbit/s modulation pulse comparison. Yellow line on RF field spy is from the RF PWM transmitter, red line is from the simple polar transmitter.

### 4.3. ISO14443 Type A High Bitrate Modulation Pulse Characteristic

Figure 17 shows simulation results for ISO14443 type A 848 kbit/s obtained with the same setup as the previous figures.

Simple polar transmitter is again unable to reach the overshoot requirement. Tests for modulation pulse shapes for bitrates higher than 106 kbit/s are not part of the current specification version; however, they may be added to future versions. The type B pulse shape requirements for bitrates higher than 106 kbit/s scale relatively linearly; however, for type A, the depth and length of modulation pulse changes with higher bitrates. This effect is the worst with type A 848 kbit/s modulation. The biggest problem lies in the fact that the rising and falling edges are often not separated by a constant amplitude part, like in type B. This situation arises when the PICC is far away and weakly coupled to the reader antenna, thus increasing the reader antenna Q and slowing the envelope. When this occurs, the length of the pulse that the RF field spy sees can become significantly different from the length that the modulator creates. This makes reaching the pulse length time requirement with a single wave shape setting at all the PICC locations specified by EMVCo and without changing the reader antenna circuit nigh impossible. One solution to this would be to measure the antenna load and adjust the wave shape to compensate. This solution was not implemented in this work, as the current version of specification only requires testing for type A and B at a bitrate of 106 kbit/s.

### 4.4. Comparison with Previous Works

A comparison with previous works is shown in Table 2. Added contribution of this work is modulation pulse wave shaping, which enables compliance with the newest specifications.

**Table 2.** Comparison with previous works.

|  | This Work | [17] | [21] |
|---|---|---|---|
| Architecture | Outphasing | Polar | Outphasing |
| Technology node | 40 nm CMOS | 180 nm CMOS | 180 nm CMOS |
| Current consumption | 3 mA@2.5 V | Not provided | 3.2 mA |
| Multi-standard support | Yes | Yes | Yes |
| Scalable power with D-class amplifiers | Yes | No | Yes |
| Adjustable ASK modulation index | Yes | No | Yes |
| Allows modulation pulse wave shaping | Yes | No | No |

## 5. Conclusions

In this work, a novel outphasing transmitter for 13.56 MHz RFID readers and NFC controllers is presented. The system is based on a differential 6-bit DTC phase modulator implemented with 2.5 V transistors in a 40 nm CMOS process, which is configured to allow amplitude modulation and expands on the previous work [17,21] to comply with the new version of EMVCo and NFC Forum specifications. Modulation bandwidth of the system is much greater than what is required by the communication protocol, which allows it to not only perform the modulation but also control the rising and falling edge shape of the modulation pulses. Output linearity of 0.2 bits was achieved over process corners, temperature, and supply voltage variations. Power consumption of the system is under 7.5 mW in the worst corner, which is negligible compared to output power of most HF RFID readers. The presented system was simulated in conjunction with an ideal D-class amplifier to test modulation performance. EMVCo and NFC Forum modulation pulse wave shape specifications are easily satisfied. In the current specification release testing for higher bitrate modulation pulse wave shapes is not required, but we tested our solution for that as well; however, a single wave shape to satisfy the requirement in all PICC positions was not found. In future work adaptive wave shaping based on antenna load could be added, which would ease the satisfaction of possible new requirements.

**Author Contributions:** Conceptualization, Ž.K. and N.S.; methodology, Ž.K.; validation, Ž.K.; formal analysis, N.S.; investigation, Ž.K.; data curation, Ž.K.; writing—original draft preparation, Ž.K. and N.S.; writing—review and editing, Ž.K., N.S. and A.P.; visualization, Ž.K.; supervision, A.P.; project administration, A.P. All authors have read and agreed to the published version of the manuscript.

**Funding:** This research received no external funding.

**Data Availability Statement:** Data is contained within the article.

**Acknowledgments:** The authors would like to thank staff members at STMicroelectronics d.o.o., Ljubljana, for helpful advice and support. We would additionally like to thank Tilen Svete for help with manuscript review and editing.

**Conflicts of Interest:** The authors declare no conflict of interest.

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
