# Peer review of "Design of Multi Standard Near Field Communication Outphasing Transmitter with Modulation Wave Shaping"

_electronics, doi:10.3390/electronics10020188_

Round 1
Author Response
Thank you very much for your careful review of the paper. The file, including the answers, is attached.
Kind regards,
Anton Pleteršek

Reviewer 2 Report
Please check the following issues:
Line 46: plural from consortium is consortia
Figures 1, 3, 4, 5 are taken from the external sources. Do the authors get the rights to use them in the paper? Additionally, the pictures have different styles so they do not look good together with the other figures.
Symbols used in Fig. 1 are not explained.
Line 58: „Modulation pulses are measured by RF field spy…” Can you explain the meaning and some details of „RF field spy”?
Line 69: Abbreviations HLZ and LLZ are not explained.
Lines 70/71: The introduction ends with the phrase „Satisfying the requirements with a modulator that does not enable active pulse shaping is extremely difficult.” It hangs up the reader with the question: what's next? Please, add a “local” summary of the introduction that explains what and why the authors did and want to present in this paper.
Line 81 Section „2. Existing transmitter solutions for near field devices”. The introduction to the section (at least one-two phrases) is missing.
Line 88: “The type select signal” – “the” have to be used to distinguish a noun from a verb.
Line 89: “When the type select is high,…”
(just two examples of the missing article) Please carefully read the paper and check the missing articles and the other linguistic and grammatical errors.
Line 107: “2.2 Multistandard Pulse Width Modulated (PWM) transmitter based on Delay Locked Loop (DLL) Architecture [20]” I don't know if it's a good idea to use a reference in a subsection name.
Line 109 “This work presents an NFC outphasing PWM [17] transmitter based…” Do you mean work [20] or your work (i.e. this work)?
Line 148: “Figure 6 shows a simulation…”. Because this is in the section entitled “Existing transmitter solutions for near field devices” the reader does not know if the simulation was made by the authors or is taken from the literature.
Figure 7: Outhph1 and Outph2 are strange abbreviations.
Figure 8: What is “PC+CP”? The line from this box is thicker than the other lines. And it goes as an upper line of the box “Current controller tapped delay line”. Is this intentional?
Figure 8 is not clear at all. Can you redraw it and make more accurate?
In Fig. 8 you use word “muxes” in the main text “multiplexers”. Please use the same word (In my opinion the “multiplexer” is better than the “mux” when it is not used as a small graphics box).
Where is the DLL in Figures 7 and 8? Can you describe the R-C-R circuit in the right part of Fig. 8? It is a pi low-pass filter?
Line 194: “The cells can also be made inverting…” – this is not clear.
Figure 9: Why the letters (a), (b), (c) are so big?
The bulk connection in the MOSFET symbol is intentionally moved from the center? It is typically symmetrically placed between D and S. Are they connected to Vdd or Vss or to the sources of transistors in your circuit?
Voltage supplies (bar and triangle symbols) are not described.
Line 211: “This drawback of the current controlled delay cells” – again an article is missing.
Line 216: “…voltage by means of current mirroring” – missing dot at the end of a phrase.
What does the „V” symbol at the top of Fig. 10?
In the text you write about two intervals: (0.8 V – 1.4 V) and (0.95 V – 1.8 V), but markers in figure are placed at 0.9V and 1.4V. Why? Additionally, in fact the value 1.8V is not depicted in Fig. 10.
Line 231: “The delay line is connected in such a way, that the 180° delayed clock is wrapped back to the differential input, as displayed in Figure 11.” Why?
The thick line in Fig. 11 is not clear. Where it is connected?
Abbreviation DFF is used in Fig. 8, but explained for Fig. 12 only.
Line 239: “as shown in Figure 12.Figure 1” typing error?
Figure 14: “Synchornization” – typo. There are no connections between the Tapped DLL and 1st, 2nd (not 2st!) and 3rd stage. Why?
Line 303: “which is over 1 W for most devices on the market today.” – a reference needed.
Line 315: “Gain and offset errors are caused by the architecture, which was optimized for area.” Could you add some details. There are no information about the chip area of the proposed transmitter.
The x axis of Fig. 15 is continuous (even with notations 7.0, 14.0 and so on), plots are continuous but they describe values in cycles. Cycles are from the discrete domain.
The values on y axes in Fig. 15 are strange, e.g. 0.788, 39.0. Why?
Lines 334 – 338 “Devices that do not address this issue can exhibit a dead zone in communication”. Is the dead zone always equal to 0? Is it possible to get the zero distance to the antenna in practice?
Line 369: “may be added to future versions.,” another typo.
Line 367: “This effect is worst with type A 848 kbit/s”.
4.4 Comparison with previous works
Your comparison takes into account the functional features only. Can you add some more detailed features as e.g. the power consumption, area size (transistor count, technology size), working scale, nonlinearity?
Line 389: “The system expands on previous work” – which work? (work of the authors or from the literature)?
Conclusions are in fact not conclusions, but an abstract. Why do you describe in conclusions how your transmitter works? The reader knows it from the paper. Please rewrite this section.
One additional remark:
A word “wave” that is used in the title is then used in the main text just one time in line 103: “No research has been done up to date in using automatic antenna tuning to control modulation wave shape. Does it describe your work really well?
Author Response
Thank you very much for your careful review of the manuscript. The file with answers is attached.
Kind regards,
Anton Pleteršek
